# Chronic Intermittent Hypoxia during Sleep Causes Browning of Interscapular Adipose Tissue Accompanied by Local Insulin Resistance in Mice

**DOI:** 10.3390/ijms232415462

**Published:** 2022-12-07

**Authors:** Tehila Dahan, Shahd Nassar, Olga Yajuk, Eliana Steinberg, Ofra Benny, Nathalie Abudi, Inbar Plaschkes, Hadar Benyamini, David Gozal, Rinat Abramovitch, Alex Gileles-Hillel

**Affiliations:** 1The Wohl Institute for Translational Medicine, Hadassah Medical Center, Jerusalem 91120, Israel; 2Faculty of Medicine, The Hebrew University of Jerusalem, Jerusalem 91904, Israel; 3The Institute for Drug Research, The School of Pharmacy, Faculty of Medicine, The Hebrew University of Jerusalem, Jerusalem 91904, Israel; 4Info-CORE, Bioinformatics Unit of the I-CORE, The Hebrew University of Jerusalem, Jerusalem 91904, Israel; 5Division of Pediatric Pulmonology, Allergy and Immunology, Comprehensive Sleep Medicine Center, Department of Child Health and Child Health Research Institute, MU Children’s Hospital, University of Missouri School of Medicine, Columbia, MO 65201, USA; 6Pediatric Pulmonology and Sleep Unit, Department of Pediatrics, Hadassah Medical Center, Jerusalem 91120, Israel

**Keywords:** hypoxia, sleep apnea, intermittent hypoxia, glucose tolerance, insulin sensitivity, brown adipose tissue, metabolism

## Abstract

Obstructive sleep apnea (OSA) is a highly prevalent condition, characterized by intermittent hypoxia (IH), sleep disruption, and altered autonomic nervous system function. OSA has been independently associated with dyslipidemia, insulin resistance, and metabolic syndrome. Brown adipose tissue (BAT) has been suggested as a modulator of systemic glucose tolerance through adaptive thermogenesis. Reductions in BAT mass have been associated with obesity and metabolic syndrome. No studies have systematically characterized the effects of chronic IH on BAT. Thus, we aimed to delineate IH effects on BAT and concomitant metabolic changes. C57BL/6J 8-week-old male mice were randomly assigned to IH during sleep (alternating 90 s cycles of 6.5% F_I_O_2_ followed by 21% F_I_O_2_) or normoxia (room air, RA) for 10 weeks. Mice were subjected to glucose tolerance testing and ^18^F-FDG PET–MRI towards the end of the exposures followed by BAT tissues analyses for morphological and global transcriptomic changes. Animals exposed to IH were glucose intolerant despite lower total body weight and adiposity. BAT tissues in IH-exposed mice demonstrated characteristic changes associated with “browning”—smaller lipids, increased vascularity, and a trend towards higher protein levels of UCP1. Conversely, mitochondrial DNA content and protein levels of respiratory chain complex III were reduced. Pro-inflammatory macrophages were more abundant in IH-exposed BAT. Transcriptomic analysis revealed increases in fatty acid oxidation and oxidative stress pathways in IH-exposed BAT, along with a reduction in pathways related to myogenesis, hypoxia, and IL-4 anti-inflammatory response. Functionally, IH-exposed BAT demonstrated reduced absorption of glucose on PET scans and reduced phosphorylation of AKT in response to insulin. Current studies provide initial evidence for the presence of a maladaptive response of interscapular BAT in response to chronic IH mimicking OSA, resulting in a paradoxical divergence, namely, BAT browning but tissue-specific and systemic insulin resistance. We postulate that oxidative stress, mitochondrial dysfunction, and inflammation may underlie these dichotomous outcomes in BAT.

## 1. Introduction

Obstructive sleep apnea (OSA) is a highly prevalent disease in children and adults, characterized by increased collapsibility of the upper airway, particularly during sleep, leading to intermittent hypoxia (IH) and hypercapnia, and increased respiratory efforts promoting sleep disruption and autonomic nervous system deregulation [1]. OSA has been independently associated with an increased risk of multiple morbidities, including atherogenesis, systemic hypertension and other cardiovascular pathologies, oncogenesis and cancer progression, cognitive and mood impairments, dyslipidemia, insulin resistance, and metabolic syndrome [2,3,4,5,6,7,8]. However, the mechanisms contributing to the increased propensity to develop insulin resistance in the context of OSA remain unclear [9,10,11,12].

Local tissue hypoxia has been hypothesized as a contributing factor to the development of visceral white adipose tissue (vWAT) inflammation and whole-body insulin resistance in obese people [13,14,15,16]. However, this assumption has been challenged [17,18,19], such that it remains unclear whether hypoxia is a cause or a consequence of obesity-associated metabolic disorders. Obesity is accompanied by adipocyte enlargement that increases inter-capillary distance, thereby markedly decreasing blood perfusion to each adipocyte and inducing an overall decline in adipose tissue oxygen tension. Notwithstanding, larger adipocytes may also have lower metabolic demands and may not necessarily promote the emergence of vWAT hypoxia [20]. Furthermore, the specific patterning of hypoxia in adipose tissues is unknown. Sustained hypoxic (SH) exposures in humans, such as those occurring in high-altitude inhabitants, increase systemic insulin sensitivity and improve glucose metabolism [21]. Interestingly, even a moderate increase in living altitude and the concomitant reduction in the atmospheric oxygen partial pressure is associated with a reduced prevalence of diabetes and obesity [22]. In contrast, intermittent exposures to hypoxia (IH), such as occurring in patients with OSA or individuals intermittently working at high altitudes [23], promotes the emergence of reduced insulin sensitivity. Indeed, despite the identical magnitude of experimental IH and SH exposures in mice (i.e., mean equivalent environmental inspired oxygen fraction (F_I_O_2_) exposures), only IH elicits insulin resistance in vWAT, which is accompanied by increased inflammation and vascular rarefaction, as well as whitening of the visceral fat, similar to the effects seen in obesity and despite weight loss [10,24].

Brown adipose tissue (BAT) is a specialized tissue with distinct anatomical characteristics and developmental origins and has been suggested as a modulator of systemic glucose tolerance through adaptive thermogenesis [25,26]. Reductions in BAT mass have been associated with obesity and metabolic syndrome [27]. Intriguingly, the role of hypoxia in BAT function has received little attention to date. In analogy with vWAT responses, the deposition of large lipid droplets in the BAT of obese subjects would be expected to lead to BAT hypoxia, and recent evidence supports this assumption [28,29]. However, since no studies have systematically characterized the effects of chronic IH on BAT, we aimed to delineate the responses of BAT to IH and examine the metabolic changes accompanying such responses.

## 2. Results

### 2.1. Body Composition and Food Consumption

Chronic exposures to IH resulted in final lower body weight (27.4 ± 0.3 g vs. 29.3 ± 0.5 g, *p* < 0.01) compared with RA. Weight gain in IH was reduced during the first two weeks of exposure, whereafter IH animals regained their weight gain patterns and tracked the weight gain trajectories of RA controls (Figure 1A). Total body adiposity as measured by T1-weighted coronal MRI images was ~50% lower in IH, compared with RA (Figure 1B), although the spatial pattern of lipolysis was not uniform—IH animals lost preferentially subcutaneous fat while preserving vWAT (Figure 1C). Accordingly, vWAT and BAT wet weights upon completion of exposures were similar between the groups (Figure 1D,E). Food consumption remained stable and did not differ when measured every week between the exposure groups.

### 2.2. Glucose Tolerance Testing

Fasting glucose levels were reduced in the IH-exposed mice compared with RA (74.9 ± 21 vs. 103.8 ± 39.1, *p* < 0.01). In line with prior studies [30,31], when presented with a glycemic challenge such as GTT, IH-exposed animals had higher glucose AUC (vs. RA: *p* = 0.01; Figure 2), indicating impaired glycemic responses in IH.

### 2.3. Brown Adipose Tissue Composition

As mentioned above, BAT total weight at the end of the exposures to IH was similar to BAT weight in RA-exposed mice. Morphologically, BAT after IH exposures was characterized by smaller lipid droplets and denser structure (Figure 3A,D,E). UCP1 protein content in BAT tissue, a marker of functional brown adipose tissue, revealed directionally higher but not statistically significant, levels of UCP1 in IH-exposed animals (Figure 3B,C). Vascularity, a phenotypic feature of a healthy and functional BAT, was increased in IH as measured by CD31 immunostaining (Figure 3A middle row). Despite these phenotypically “brown” features, the levels of P2RX5, a marker of classical brown adipose tissue, were reduced (Figure 3F).

As mitochondria play an important role in BAT function, we analyzed mitochondrial electron transport chain (ETC) protein levels and the amount of DNA content. We found that in BAT harvested from IH-exposed mice, protein levels of ETC complex III were reduced, whereas complex I was increased (Figure 3G,H). In addition, IH resulted in a non-significant reduction in BAT-mtDNA content (Figure 3I).

Bone-marrow-derived pro-inflammatory Ly6C^high^ macrophage infiltration has been associated with obesity and dysfunction of brown adipose tissue [32]. FACS analysis of the BAT macrophage population revealed no differences between RA and IH in the proportion of CD11b^+^F4/80^+^ macrophages out of total cells (1.2% vs. 0.9%, *p* = 0.354), or the proportion of resident CD64^+^/Ly6C^low^ cells (79.5% vs. 77.2%, *p* = 0.863). However, the Ly6C^high^ population was significantly increased in IH (4% vs. 2.5%, *p* < 0.001, Figure 4)

### 2.4. Brown Adipose Tissue Transcriptional Profile

Next, we examined how chronic IH exposures affect gene expression profiles in BAT. Following chronic IH exposures for 10 weeks, BAT tissue demonstrated a differential activated transcriptional profile with some changes mirroring the morphological changes described above.

GSEA pathway analysis indicated that the pathways that were most upregulated in IH compared with RA, were related to oxidative phosphorylation, adipogenesis, fatty acid oxidation, reactive oxygen species, and DNA repair (Figure 5A,B). The most suppressed pathways in IH compared with RA, were hypoxia, myogenesis, angiogenesis, and some inflammatory signaling such as the IL-4 and IL-13 pathways. We further validated the RNA-seq results with RT-PCR analysis for gene expression of specific targets relevant to BAT biology. IH BAT tissues were characterized by increased transcription of genes related to thermogenesis (UCP1, Dio2, PGC1a), compared with RA controls (Figure 5C).

### 2.5. Brown Adipose Tissue Glucose Metabolism

Finally, given the morphological and transcriptional differences in BAT between the experimental conditions, we examined whether they result in functional differences in glucose utilization: PET–MRI analysis revealed a ~23% reduction in the total absorption of ^18^F-FDG following overnight fasting in total body fat and BAT tissues of IH-exposed mice when compared with control mice (Figure 6B). As brown adipocytes and muscle share a common developmental origin [33], we also examined the effect of IH exposures on glucose uptake in skeletal muscle—a similar ~30% reduction was observed in skeletal muscle.

Next, we examined whether the differential glucose absorption in BAT following IH exposures is accompanied by alterations in insulin signaling—a subgroup of mice (*n* = 3–5/group) was injected with insulin i.p. following overnight fasting and sacrificed 5 min later. We then analyzed AKT phosphorylation in the BAT of those mice. IH-exposed BAT demonstrated a 69% reduction in AKT phosphorylation in response to insulin (Figure 6C,D). As with PET–MRI findings, skeletal muscle demonstrated a similar trend (Appendix A). Taken together, these data suggest a peripheral (BAT and muscle) resistance to insulin signaling following IH.

## 3. Discussion

The current study aimed to assess morphologic, transcriptomic, and functional differences in the BAT responses in mice exposed to chronic IH, similar to the episodic oxygenation changes that characterize OSA and many other respiratory diseases. IH-exposed mice demonstrated systemic glucose intolerance and manifested BAT and muscle insulin resistance. BAT transcriptomic analysis following chronic IH exposures revealed the activation of pathways related to fatty acid oxidation and oxidative phosphorylation, potentially indicating a compensatory mechanism due to reduced glucose metabolism. In addition, oxidative stress pathways were upregulated. Morphologically, BAT in IH mice was skewed towards a more brown phenotype, with increased vascular density, smaller lipid droplets, and directionally higher UCP1 expression. Conversely, mitochondrial DNA content was reduced in BAT following IH, as was the amount of ETC complex III protein, accompanied by Ly6C^high^ macrophage infiltration.

A large body of evidence, including our prior work, has shown that chronic IH exposures mimicking OSA in mice induce visceral adipose tissue dysfunction, with abnormal adipokine secretion and lipid metabolism, along with the presence of increased local and systemic inflammation, and ultimately insulin resistance [10,30,34,35,36]. The current study recapitulates some of the previously described metabolic alterations under chronic IH and adds BAT to the list of metabolically important tissues affected by chronic episodic hypoxia.

BAT an important organ that functions as a “metabolic sink” that can metabolize excess nutrients through oxidation and thermogenesis [37]. BAT is unique in its ability to convert chemical energy directly into heat upon sympathetic stimulation, thus, increasing caloric consumption and promoting metabolic health. Accordingly, sympathetic stimuli, such as cold exposures or administration of β-adrenergic agonists lead to increases in the amount of BAT. Conversely, increases in the amount of vWAT and reduction in BAT depots have been associated with metabolic syndrome and obesity. In the context of the complexity of adipose tissue bioenergetics, brown adipocytes can develop within vWAT upon sympathetic stimulation, so-called “beige” fat [38], and reflect an expansion of classical BAT tissues. In recent years, BAT has gained a renewed interest with the unequivocal demonstration of active and dynamic depots in human adults and its role in insulin sensitivity in humans [27,39,40]. Thus, both brown and beige adipocytes are now appealing targets to increase energy expenditure and combat obesity [41].

In the context of hypoxia, both sustained and intermittent hypoxia lead to increased catecholaminergic release and enhanced sympathetic activation [42,43,44,45]. Based on such responses, one would have expected an increase in BAT mass along with increased browning of vWAT, all of which would have been anticipated to enhance insulin sensitivity. However as shown herein, IH elicited reduced insulin sensitivity of BAT despite the increased browning of BAT after such prolonged IH exposures. These findings suggest a dichotomous and somewhat paradoxical response to IH by BAT that requires further exploration. Furthermore, chronic IH, such as found in patients with OSA, promotes whitening of the visceral adipose depot with concomitant insulin resistance and vascular rarefaction rather than the expected browning predicted by increased sympathetic activity and catecholamine levels. These observations, therefore, suggest that the presentation of hypoxia and its duration may yield markedly different phenotypic responses that are specific to each organ or cell type [46].

Surprisingly, few studies have examined BAT changes following IH exposures. Yao et al. [47] evaluated lipid handling by different tissues under IH, and found reduced lipoprotein lipase activity in BAT of IH-exposed mice, leading to impaired serum triglyceride clearance. Martinez et al. [48] found that chronic IH reduced BAT amount, but they did not characterize the metabolic consequences of this effect. The findings of our study suggest that chronic whole-body IH promotes some aspects of the brown phenotype of BAT while simultaneously reducing whole-body and BAT-specific glucose tolerance. This seemingly paradoxical effect may be ascribable to the chronic activation of sympathetic signaling, [49] which may underlie some of the IH-associated metabolic perturbations [50]. Indeed, brown fat thermogenesis is primarily driven by the sympathetic nervous system, and β3 adrenergic receptor agonists effectively mimic cold-induced thermogenesis [51]. Increased thermogenesis and UCP1 expression have also been linked to increased fatty acid oxidation by BAT [52], similar to what is suggested by the transcriptomic findings in the IH-exposed mice in the current study. However, it has also been shown, that thermogenesis may not necessarily be associated with improved BAT nutrient handling and insulin tolerance [53] and some metabolic function of BAT is independent of UCP1-linked thermogenesis [54,55]. Thus, the peripheral insulin resistance we observed in BAT under chronic IH conditions is not in line with the brown phenotype in BAT and may stem from other pathophysiological processes.

The strong suppression of myogenesis following IH observed in the transcriptome signature deserves mention—white and brown adipose tissues are derived from different embryological origins, whereby brown preadipocytes uniquely share an overlapping transcriptional program with muscle cells [56], thereby explaining the abundance of mitochondria in both of these cell types. Conversely, white adipose tissue lacks a myogenic gene signature. Therefore, myogenesis suppression observed in IH-exposed BAT may reflect some “whitening” effect in line with the reduction in mitochondrial content.

There are several dominant contributors to the pathogenesis of dysfunctional adipose tissue in obesity: unresolved inflammation, oxidative stress, inappropriate extracellular matrix (ECM) remodeling, and insufficient angiogenic potential [57,58]. Our RNAseq data suggest that at least some of these maladaptive responses are activated in IH-exposed BAT tissue such as increased oxidative stress (due to repeated oxygenation–reoxygenation cycles and fatty acid oxidation) and reduction in the anti-inflammatory IL-4-dependent signaling, which may lead to BAT dysfunction [59,60].

Increased vascularity of adipose tissue improves both the delivery and handling of nutrients [20]. Increased BAT vascularity as occurred here in IH exposures would again be predicted to result in improved glucose handling, which was not the case. However, neovascularization in adipose tissues could intensify the inflammatory processes induced by IH by increasing the infiltration of inflammatory cells, as exemplified by the FACS data on the Ly6C^high^ macrophage infiltration.

### Limitations

Several limitations of the current work should be acknowledged. While we provide a deep analysis of BAT following IH exposure, we could not examine the functional thermogenic capacity of BAT in our experiments. As discussed, thermogenic BAT activity might not be in line with nutrient handling and metabolic function. Nonetheless, the experiments utilizing PET-imaging were performed in live animals, and at an ambient temperature (without cold stimulation usually utilized for BAT imaging [61]) and, thus, provide a true measure of glucose handling in vivo. Furthermore, the changes described in BAT mirror metabolically maladaptive phenotype in IH in other tissues, suggesting BAT dysfunction may play a role in OSA-related insulin resistance. In addition, the current study is descriptive in nature, and more definitive experiments are needed—for example, the adoptive transfer of BAT from IH-exposed animals to naïve ones may provide evidence of a causal role for BAT in IH-associated metabolic dysfunction. A study assessing thermogenic capacity under IH could provide an answer as to whether such BAT thermogenesis is truly dissociated from glucose handling following IH. Finally, sex and age, two well-recognized modulators of the metabolic response, were not addressed in the current study and deserve future investigation.

## 4. Methods and Materials

### 4.1. Animals and Hypoxic Exposures

All studies were approved by the institutional animal care and use committee of the Hebrew University of Jerusalem (AAALAC accreditation no. 1285, approval MD-20-16401-4, 15/12/2020). C57BL/6J 8-week-old male mice (ENVIGO; Indianapolis, IN, USA) were randomly assigned to IH or normoxia (room air, RA) and exposed for 10 weeks using computer-controlled environmental chambers (Oxycycler A44XO, BioSpherix, Parish, NY, USA). The pattern of IH consisted of alternating cycles of 90 s (6.5% F_I_O_2_ followed by 21% F_I_O_2_) for 12 h/d (7:00 a.m. to 7:00 p.m.), mimicking moderate to severe OSA in humans [62,63]. The control group (RA) was exposed to continuous circulating room air (21% F_I_O_2_). Body weight and food intake were recorded weekly.

### 4.2. In Vivo Micro-PET–MRI Scanning

Experiments were performed at the Wohl Institute for Translational Medicine at Hadassah Hebrew University Medical Center. PET–MRI images were acquired on a 7T 24 cm bore, cryogen-free MR scanner based on the proprietary dry magnet technology (MR Solutions, Guildford, UK) with a 3-ring PET insert that uses the latest silicon photomultiplier (SiPM) technology [64]. The PET subsystem contains twenty-four detector heads arranged in three octagons of 116 mm diameter. For MRI acquisition, a mouse quadrature RF volume coil was used. Mice were anesthetized with isoflurane and vaporized with O_2_. Isoflurane was used at 3.0% for induction and 1.0–2.0% for maintenance. To determine the distribution of [^18^F]-FDG in fasted mice, 200 μCi of a tracer was injected into the tail vein. Mice were subjected to 31 min dynamic PET scans, a homemade small catheter was inserted in the proximal tail vein and the tracer was injected after positioning the mouse in the micro-PET/MRI scanner. Mice were positioned on a heated bed, which allowed for continuous anesthesia and breathing rate monitoring. For dynamic scans, the acquired data were binned into 25 image frames (1 × 60, 6 × 10, 8 × 30, 5 × 60, and 4 × 300 s). During the PET scan acquisitions, T1 and T2-weighted coronal spin echo images were collected for anatomical evaluation. Coronal T1 weighted images were acquired using the following parameters: TR = 1100 ms, TE = 11 ms, echo spacing = 11 ms, FOV = 6 × 3 cm, slice thickness = 1 mm, 4 averages. Coronal T2 weighted images were acquired using the following parameters: TR = 4000 ms, TE = 45 ms, echo spacing = 15 ms, FOV = 6 × 3 cm, slice thickness = 1 mm, 4 average.

### 4.3. PET/MRI Image Processing

Images were analyzed using VivoQuant 4.0 pre-clinical image post-processing software (Invicro, Boston, MA, USA). PET–MRI raw data were processed using the standard software provided by the manufacturers. PET data were acquired in list mode, histogrammed by Fourier re-binning, and reconstructed using a 3D-OSEM algorithm, with standard corrections for random coincidences, system response, and physical decay applied. The PET/MR scanner-reconstructed PET images were quantitated using a system-specific ^18^F calibration factor to convert reconstructed count rates per voxel to activity concentrations (%ID/g). Manual tissue segmentation of BAT and automatic segmentation of WAT were performed on co-registered 3D MR images. The regional ROIs were then used to calculate tissue radiotracer uptake from the reconstructed PET images (Figure 6A) [64].

### 4.4. Glucose and Insulin Tolerance Tests

A glucose tolerance test was performed in overnight fasted mice by injecting D-glucose (2 g/kg body weight) intraperitoneally. Blood glucose concentrations were monitored by tail bleeding at 0, 15, 30, 60, and 120 min after the glycemic load. Blood glucose levels were measured using a glucometer (ACCU-CHEK, Roche, Indianapolis, IN, USA). Insulin levels were measured using an ultrasensitive insulin ELISA kit (#90060, Crystal Chem, Elk Grove Village, IL, USA) [65]. A subset of mice was used to examine tissue insulin sensitivity—mice were fasted for 16 h before i.p. injection of 2.5 U/kg insulin (NovoRapid, Novo Nordisk, Bagsværd, Denmark). After 5 min, mice were euthanized and tissues were harvested, shock frozen in liquid nitrogen, and stored for later assays.

### 4.5. Immunohistochemistry

Tissue samples were fixed in 4% formaldehyde for 24 h, embedded in paraffin, sectioned serially (5 μm), and stained with hematoxylin–eosin (H&E) or subjected to immunofluorescence with CD31 antibody (1:50, #ab28364, Abcam, Cambridge, UK). The secondary antibody was purchased from Jackson ImmunoResearch (#111-165-045, West Grov, PA, USA). Fluorescent images were taken on a Nikon C1 confocal microscope at an original magnification of ×40. For immunofluorescence studies, BAT samples were fixed in 4% PFA overnight at 4C, transferred to 30% (wt/vol) sucrose overnight at 4C, and embedded in OCT (#4583, Tissue-Tek, the Netherlands). Sections of 30 μm were cryosectioned and stained for natural lipids (Nile Red, #72485,7 ug/mL, Sigma Aldrich, St. Louis, MO, USA), Alexa Fluor^®^647 phalloidin (#ab176759, Invitrogen, Waltham, MA, USA), and DAPI (1:10,000, #ab228549, Abcam, Cambridge, UK).

### 4.6. Western Blot Analysis

Tissues were homogenized in lysis buffer with protease and phosphatase inhibitor cocktails (Sigma-Aldrich, MO, USA). Total protein content was measured using the Bradford assay (Bio-Rad, Hercules, CA, USA). Homogenate proteins (30 μg) were separated on 10% SDS-acrylamide gel and transferred to PVDF membranes (Amersham, Buckinghamshire, UK). Immuno-blotting was performed using primary antibodies: anti-phospho-Akt (Ser473); Akt (Cell Signaling Technology, #8200, Beverly, MA, USA), or UCP1 (1:5000, #ab10983, Abcam, Cambridge, UK). The signal was detected using a chemiluminescence detection system (#34577, Thermo Fischer Scientific, Waltham, MA, USA) according to the manufacturer’s instructions by using the ChemiDoc XRS+ and Quantity One software (Bio-Rad Laboratories, CA, USA).

### 4.7. Mitochondrial DNA Quantification

To measure mitochondrial DNA (mtDNA) copy number, DNA was extracted from BAT tissues by using the DNeasy Blood and Tissue kit (#69504, Qiagen, Hilden, Germany) according to the manufacturer’s instruction and quantified by Real-time PCR. The mtDNA copy number per genomic DNA copy number was calculated using primers for mouse mitochondria (MtDNA) and beta-2 microglobulin (B2M) [66].

### 4.8. RNA-Seq and Analysis

Total RNA was extracted from brown adipose tissue using an RNeasy lipid tissue kit (#74804, Qiagen, Hilden, Germany) according to the manufacturer’s protocols. The integrity of the RNA was confirmed using a Bioanalyzer 2100 (Agilent Technologies, Santa Clara, CA, USA). RNA-seq libraries were prepared using the KAPA Stranded mRNA-Seq Kit Illumina^®^ and sequenced using the Illumina NextSeq 500 platform to generate more than 32 million 86 bp single-end reads per sample. Raw reads were preprocessed using Cutadapt [67] and aligned to the mouse genome version GRCm38 (with genome annotations of Ensembl release 99) using TopHat [68]. Quantification was performed with HTSeq-count, with strand information set to ‘reverse’ [69]. DESeq2 analysis for the identification of differentially expressed genes was performed [70]. Pair-wise comparisons were tested with default parameters, except not using the independent filtering algorithm. To reduce the number of false-positive results, significant genes were further filtered. The criteria for filtering included both a demand for a large enough effect (log2FoldChange) and being significant, with padj < 0.1. The demand for log2FoldChange was baseMean-dependent and required a baseMean above 5 and an absolute log2FoldChange bigger than 5/sqrt (baseMean) + 0.3 (for highly expressed genes this means a requirement for a fold-change of at least 1.2, while genes with a very low expression would need a 5.8-fold change to pass the filtering).

Gene set enrichment analysis (GSEA): Whole differential expression data were subjected to gene set enrichment analysis using GSEA [71]. GSEA uses all differential expression data (cutoff independent) to determine whether a priori-defined sets of genes show statistically significant, concordant differences between two biological states. We used the following gene sets collections: Hallmark, Gene Ontology Biological Process (GOBP), Kyoto Encyclopedia of Genes and Genomes (KEGG), REACTOME, and Wiki pathways, all taken from the molecular signatures database MsigDB [72].

### 4.9. Real-Time PCR

cDNA was synthesized from 1 ug of RNA using a High-Capacity cDNA Reverse Transcription Kit (#84002 Quanta Biosciences, MA, USA). Real-time PCR was performed with SYBR Green mix (#84071 Quanta Biosciences, MA, USA). All reactions were performed in triplicate in 384-well plates using the CFX384 Real-Time System (Bio-Rad). The relative amount of mRNA was calculated using the comparative Ct method after normalization to YWHAZ/b-actin housekeeping gene levels. RT-PCR primers are described in Table 1.

### 4.10. Isolation of Stromal Vascular Fraction (SVF) and Flow Cytometry Analysis

Adipose tissue macrophages were defined as CD11b/F4/80 double-positive cells, from which resident (anti-inflammatory) and bone-marrow-derived (pro-inflammatory) macrophages were identified as CD64+ or Ly6c^high^ cells [73,74], respectively. Isolation and analysis were performed as described previously [75]. All antibodies were from Biolegend (San Diego, CA, USA).

### 4.11. Statistical Analysis

The data are presented as means and standard errors of the mean. Student’s *t*-tests or two-way analyses of variance were used to compare RA and IH groups, with post hoc tests for multiple comparisons when appropriate. For data that were not distributed normally, the nonparametric Mann–Whitney rank-sum test was utilized. Two-tailed *p* values were calculated for all pairwise multiple comparison procedures using the Student–Newman–Keuls test among groups. A *p* value of <0.05 was considered statistically significant.

## 5. Conclusions

In summary, current studies provide initial evidence for the presence of a maladaptive response of BAT in response to IH during sleep—chronic intermittent exposure to fluctuating levels of oxygen resulted in browning of interscapular BAT accompanied by BAT and systemic insulin resistance. Accordingly, chronic exposure to IH may uncouple adaptive thermogenesis from glucose tolerance in specific metabolically active tissues such as BAT, and such a phenomenon clearly deserves further investigation regarding IH stimulus magnitude, cycling, and time domain characteristics. Translationally, future studies will be needed to examine the relevance of these findings in the context of human OSA and other respiratory disorders, aiming to identify the potential mechanisms governing such divergent structural, cellular, and metabolic responses.

## Figures and Tables

**Figure 1 ijms-23-15462-f001:**
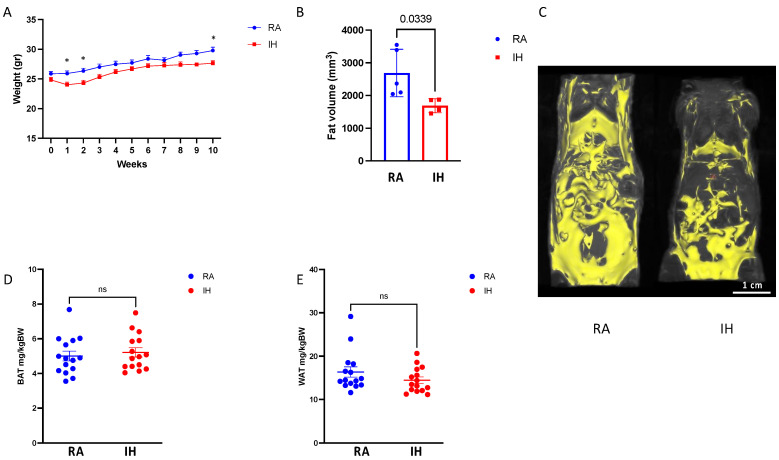
Chronic intermittent hypoxia exposure results in lower body weight and adiposity. Body composition comparison of mice exposed to chronic intermittent hypoxia or room air conditions (*n* = 14–15). (**A**) Weight gain trajectory throughout the exposure. (**B**) Whole body fat volume derived from the MRI scans (*n* = 4–5). (**C**) Representative MRI images from each of the experimental groups demonstrating reduced adiposity in IH with preservation of visceral fat (right panel). (**D**) Brown adipose tissue (BAT) weights normalized to body weight. (**E**) Visceral fat (WAT) weight at the end of exposure normalized to body weight. Data are presented as means ± SEM. Pairwise comparisons were performed using Student’s *t*-test. * *p*-value below 0.05. Weight statistics (**A**) were compared with two-way ANOVA using repeated measures and correction for multiple comparisons. ns–non-significant.

**Figure 2 ijms-23-15462-f002:**
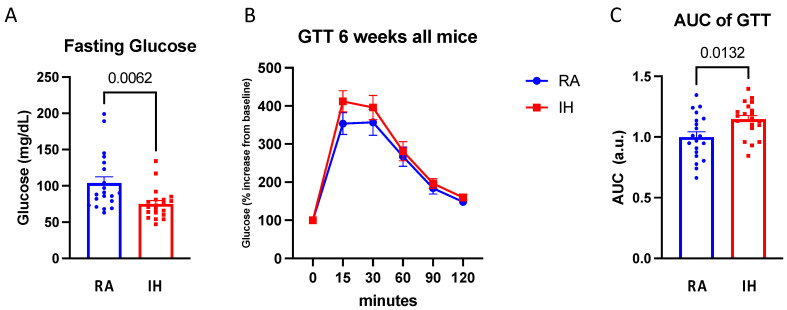
Chronic intermittent hypoxia induces systemic glucose intolerance. Fasting glucose tolerance test following 6-week exposure. (**A**) Fasting glucose levels. (**B**) Glucose curve normalized to baseline glucose levels and the corresponding. (**C**) Area under the curve (AUC) (*n* = 17–20). Data presented as means ± SEM. Pairwise comparisons were performed using Student’s *t*-test. AUC was calculated using the trapezoidal method.

**Figure 3 ijms-23-15462-f003:**
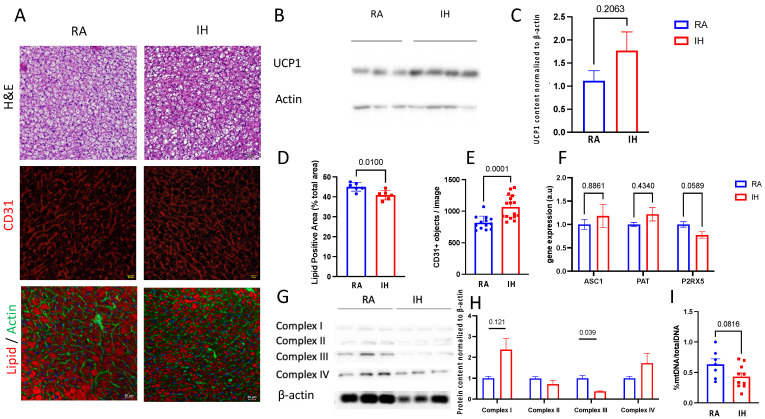
Chronic intermittent hypoxia promotes the browning of interscapular BAT with evidence for mitochondrial dysfunction. Phenotypic changes in BAT following chronic intermittent exposures compared with room air controls. (**A**) Representative images of BAT from the experimental groups: H&E (top row), vascular marker CD31 (middle row), and lipid-staining Neil Red with Actin-staining phalloidin (bottom row), bar = 20 μm. (**B**,**C**) Western blot analysis of UCP1 expression in BAT, normalized to β-actin. (**D**) Quantification of lipid positive area, and (**E**) Vessel density (5–6 randomly selected images from each slide, *n* = 3 biological replicates). (**F**) mRNA levels of 3 markers of adipose tissue phenotype ASC1 (white), PAT (beige), and P2PRX5 (brown). (**G**,**H**) Western blot analysis of the expression of mitochondrial electron transport chain proteins normalized to β-actin. (**I**) Mitochondrial DNA quantification normalized to genomic DNA amount (B2M). Data are presented as mean ± SEM. Pairwise comparisons were performed using Student’s *t*-test.

**Figure 4 ijms-23-15462-f004:**
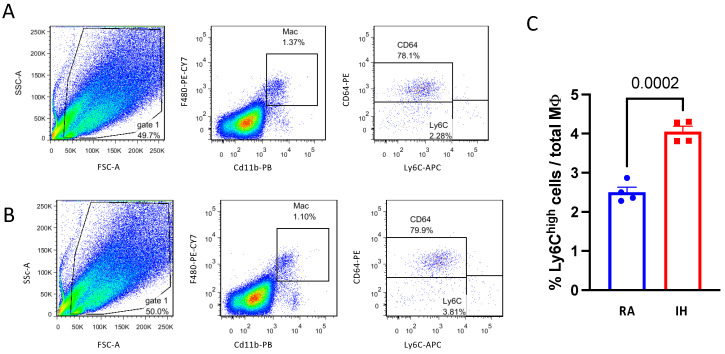
Chronic intermittent hypoxia promotes pro-inflammatory macrophage infiltration into interscapular BAT. (**A**) Representative flow cytometry images of macrophages isolated from BAT of RA or (**B**) IH mice—gated as CD11b/F480 double-positive cells (middle panel) and further stained with anti-CD64 (resident, anti-inflammatory) and anti-Ly6c (bone-marrow derived, pro-inflammatory) antibodies. (**C**) Quantification of the Ly6C^high^ macrophages in BAT demonstrated an increased percentage in IH. Note that the majority of CD64+ macrophages were Ly6C^low^ and vice versa. The percentage of macrophages and the CD64+ subpopulation was not different between the experimental conditions (*n* = 4). Pairwise comparisons were performed using Student’s *t*-test. MΦ—macrophage.

**Figure 5 ijms-23-15462-f005:**
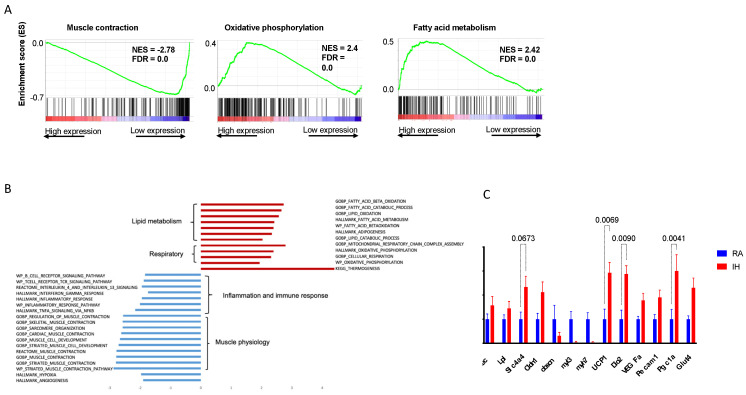
Chronic intermittent hypoxia leads to global transcriptomic changes in BAT. (**A**) Example of pathways up/downregulated in IH-exposed BAT. (**B**) The most altered pathways between IH and room air-exposed BAT. (**C**) Expression of selected genes in signaling pathways relevant to brown adipose tissue (left to right): lipid droplet structure (Cidec, Lpl, Slc4a4, Cldn1), myogenesis (obscn, myl3, myh7), thermogenesis (UCP1, Dio2), angiogenesis (Vegfa, Pecam1, PGC1a), and glucose utilization (Glut4).

**Figure 6 ijms-23-15462-f006:**
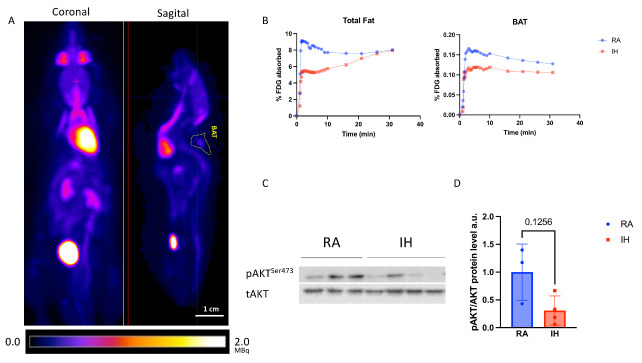
Chronic intermittent hypoxia promotes BAT-specific insulin resistance. (**A**) Representative ^18^F-Fluorodeoxyglucose (FDG) absorption images demonstrating the anatomical location of BAT. (**B**) Quantification of the ^18^F-FDG consumption in total fat, demonstrating reduction in IH, compared with RA in total body fat (left panel) and BAT (right panel) (*n* = 3–5). (**C**,**D**) Western blot analysis of phosphorylated AKT(Ser473) protein levels normalized to total AKT in BAT following i.p. insulin stimulation (*n* = 3–4).

**Table 1 ijms-23-15462-t001:** Primers used for RT-qPCR analysis.

Target	Forward	Reverse
GLUT4	GTCGGGTTTCCAGCAGATC	AAACTGAAGGGAGCCAAGC
PGC1a	AAATCATATCCAACCAGTACA	CATCTGTCAGTGCATCAAAT
Myh7	TGCTGTTTCCTTACTTGCTA	GGATTCTCAAACGTGTCTAGT
Cldn1	TACAGTGCAAAGTCTTCGACT	GACACAAAGATTGCGATCAG
Slc4a4	GATGAAGCTGTCCTGGACA	GACCCCAATGTAGATCGTG
Lpl	GTTTGGCTCCAGAGTTTGAC	CAAGTGTCCTCAGCTGTGTCT
Cidec	CTCACAGCTTGGAGGACCT	CAGGGCTTGGAAGTATTCTT
Myl3	TGATGCCTCCAAGATTAAG	CGTATGTGATCTTCATCTCG
YWHAZ	AGAAGATCGAGACGGAGCT	GCCAAGTAACGGTAGTAGTCA

Act (qMmuCED0027505), UCP1 (qMmuCID0005832), and DIO2 (qMmuCEP0052679) were purchased as gene expression assays from BIO-RAD.

## Data Availability

The data presented in this study are available on request from the corresponding author. The data are not publicly available as they contain results of an ongoing work yet to be published.

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
