# Peer review of "Chronic Intermittent Hypoxia during Sleep Causes Browning of Interscapular Adipose Tissue Accompanied by Local Insulin Resistance in Mice"

_ijms, 2022, doi:10.3390/ijms232415462_

Round 1
Reviewer 1 Report
Thank you for giving the reviewer an opportunity to review a well written scientific manuscript.
The article is devoted to the evaluation of intermittent hypoxia during sleep (alternating 90 s cycles of 6.5% FIO2 followed 28 by 21% FIO2) in C57BL/6J 8-week-old 27 male mice for 10 weeks, especially focusing on metabolic, inflammatory panels, and mitochondrial contents brown adipose tissue.
The article is of scientific interest. The authors have clearly elaborated the results and methods. It is written in correct scientific language.
The results of this study demonstrates glucose intolerance despite reduced body and fat mass in BAT.
BAT of IH mice showed smaller lipids, increased vascularity, and a trend towards higher protein levels of UCP1.
However, mitochondrial DNA content and protein levels of respiratory chain complex III were reduced.
Pro-inflammatory macrophages were more abundant in IH-exposed BAT.
Increases in fatty acid oxidation and oxidative stress pathways in IH-exposed BAT, along with a reduction of pathways related to myogenesis, hypoxia, and IL-4 anti-inflammatory response in BAT.
However, the reviewer would like to point out and few things, and ask the authors to clarify some point in the following;
1. Figure 6 B shows void in the peer review files. This needs to be clarified, and match with the text in the results section (page 7 line 186).
; PET-MRI analysis revealed a ~23% reduction in the total absorption of 186 18F-FDG following overnight fasting in total body fat and BAT tissues of IH-exposed mice 187 when compared to control mice (Fig 6B).
2. The exclusive role of BAT differing from vWAT or other fat tissues in the subcutaneous area lies in its thermogenic potential. Although the authors have demonstrated the structural, and molecular (various proteins and transciptional factors) changes in BAT following IH, the lack of thermogenic potential evaluation seriously limits the implication and translational merits in this study. The authors should more emphasize on this aspect, and would be able to suggest a future study model regarding this issue.
3. In addition, it would be an area of interest if there is any differeces between BAT, vWAT, and subcutaneous adipose tissues upon exposure to IH, and whether these differences exists in the persistent hypoxia, and control groups.
Reviewer 2 Report
Comments
This manuscript by Dahan et al. 2022 submitted in the of International Journal of Molecular Sciences (ijms-1960229) describes “Chronic Intermittent Hypoxia During Sleep Causes Browning of Interscapular Adipose Tissue Accompanied By Local Insulin Resistance In Mice”. The authors provide initial evidence for the presence of a maladaptive response of interscapular BAT in response to chronic IH mimicking OSA, resulting in a paradoxical divergence, namely BAT browning but tissue-specific and systemic insulin resistance. The authors postulate that oxidative stress, mitochondrial dysfunction, and inflammation may underlie these dichotomous outcomes in BAT. The manuscript is well written and easy to read. However, while this is a worthwhile study, I have a number of concerns, which need to be addressed.
General comments
1. Please provide the catalog number for all reagents and material used.
2. Scale bars, axes, and significant bars should be equalized in weight and size to match in all Figures. Please label all X axis with the appropriate group name and use stars for the significance instead of numbers.
3. The authors should provide detailed statistical analysis and method used either in the results or in the figure caption.
4. The manuscript should be proofread to fix typo errors, extra spaces, periods, and use Font Symbol where necessary.
Majors
1. There are number of inconsistences between the description of the results and the actual data they provided.
2. Figure 5B have missing data (it is an empty figure).
3. Number of animals is too low in some experiments to support the conclusion of the study.
4. Please provide your gating strategy for FACS.
Minors
1. Number of animals is low (n 3-4/group) in some experiments, did the authors calculate the power analysis and the number of animals required for each experiment beforehand? If yes, please provide the results. If not, please justify it.
2. Line 91-92: add Figure number.
3. Figure 1A: please add the significance stars where the difference is.
4. Figure 1B: please use a star for the significance.
5. Figure 1D-E: Please label X axis.
6. Please add Figure for food intake.
7. Line 109: please rename this section as Glucose tolerance test.
8. While the AUC did show a difference between groups (Fig 2C), there is no significance difference in GTT between groups (Fig 2B). How the authors interpret their results?
9. Line 120-123: please move first sentence to Introduction or Discussion and the second sentence to section 2.1.
10. Line 124-126: base on the p, this is not a trend at all. Therefore, this sentence should be rephrased.
11. Fig 3D-E description is missing in the Results (section 2.3).
12. Fig 5A-B: increase the size of the figure, it is hard to see, especially Fig 5B.
13. Fig 6A should show both RA and IH to match the description in section 2.5.
14. Line 190: please add the data in Supplemental material.
15. Fig 5B description is missing in the Results (section 2.5).
Reviewer 3 Report
The authors studied the chronic effects of intermittent hypoxia (IH) on brown adipose tissue (BAT) and other metabolic factors. They found that IH led to maladaptive responses, such as glucose intolerance, browning of BAT, altered gene expression in BAT, and insulin resistance.
Overall, the manuscript is well-written with coherent logic and clear presentation of findings. The findings are interesting and potentially clinically relevant. I have some relatively minor concerns:
1. The first sentence of the introduction needs to be supported by references. Please add relevant citations.
2. Lines 92-94 ("Weight gain... of RA controls"): The described pattern does not seem very clear in Fig 1. Could the authors include some statistics to support it? Besides, the error bars were only shown for the RA group. It would be great to include the error bars for the IH group as well.
3. In Fig 1a, it is unclear what the bracket (with 0.0042) was referring to. It seems that its location has shifted. Please double-check.
4. Line 113 ("in line with prior studies, ..."). It would be great if the authors can include the citations of the prior studies.
5. In Fig 2A and 2C, error bars are missing for the IH group.
6. Line 138-139 ("In addition..."). The p-value is >0.05, so it shouldn't be considered a significant change.
7. Figure 6A was not referenced in the main text. Please make sure to cite it in the relevant text.
8. Figures 6B-D do not seem to be shown correctly. Please double-check.
Author Response
Thank you for your kind feedback.
Please see below our point-by-point response:
The authors studied the chronic effects of intermittent hypoxia (IH) on brown adipose tissue (BAT) and other metabolic factors. They found that IH led to maladaptive responses, such as glucose intolerance, browning of BAT, altered gene expression in BAT, and insulin resistance.
Overall, the manuscript is well-written with coherent logic and clear presentation of findings. The findings are interesting and potentially clinically relevant. I have some relatively minor concerns:
Response: Thank you for your favorable review and instrumental feedback.
1. The first sentence of the introduction needs to be supported by references. Please add relevant citations.
Response: added.
2. Lines 92-94 ("Weight gain... of RA controls"): The described pattern does not seem very clear in Fig 1. Could the authors include some statistics to support it? Besides, the error bars were only shown for the RA group. It would be great to include the error bars for the IH group as well.
Response: We have performed additional analysis and added asterisks to figure 1 to denote those differences most prominent between the two experimental groups. As can be seen, when correcting for multiple comparisons and repeated measures, the difference between RA and IH groups is most prominent in the first two weeks of exposure (and in the final weights). Error bars were corrected.
3. In Fig 1a, it is unclear what the bracket (with 0.0042) was referring to. It seems that its location has shifted. Please double-check.
Response: This was added per review 2 requests to mirror the text describing lower final weight in IH vs RA mice. We changed the figure now to make it more clear.
4. Line 113 ("in line with prior studies, ..."). It would be great if the authors can include the citations of the prior studies.
Response: Thank you - we added some references.
5. In Fig 2A and 2C, error bars are missing for the IH group.
Response: We corrected this.
6. Line 138-139 ("In addition..."). The p-value is >0.05, so it shouldn't be considered a significant change.
Response: Agreed - we have added "a non-significant" to the text.
7. Figure 6A was not referenced in the main text. Please make sure to cite it in the relevant text.
Response: Thank you for pointing this out. We added this to the methods section describing the ROI chosen for BAT analysis.
8. Figures 6B-D do not seem to be shown correctly. Please double-check.
Response: To avoid confusion we added a more detailed description of subfigure B (total fat left panel, BAT right panel).
Round 2
Reviewer 1 Report
The authors have well revised and re-organized the manuscript, thereby improving the quality of the article.
Author Response
Thank you
Reviewer 2 Report
Thank you for taking the time to address my comments. Unfortunately, I cannot accept the manuscript in its present form.
Author Response
This is unfortunate and we defer to AE for final decision